# Influence of Single Dose Enrofloxacin Injection on Development of Fluoroquinolone Resistance in *Campylobacter jejuni* in Calves

**DOI:** 10.3390/antibiotics11101407

**Published:** 2022-10-13

**Authors:** Debora Brito Goulart, Ashenafi Feyisa Beyi, Zuowei Wu, Mehmet Cemal Adiguzel, Samantha Wilson, Changyun Xu, Jinji Pang, Renee Dewell, Grant A. Dewell, Paul J. Plummer, Qijing Zhang, Orhan Sahin

**Affiliations:** 1Department of Veterinary Microbiology and Preventive Medicine, College of Veterinary Medicine, Iowa State University, Ames, IA 50011, USA; 2Center for Food Security and Public Health, College of Veterinary Medicine, Iowa State University, Ames, IA 50011, USA; 3Department of Veterinary Diagnostic and Production Animal Medicine, College of Veterinary Medicine, Iowa State University, Ames, IA 50011, USA; 4National Institute of Antimicrobial Resistance Research and Education, Iowa State University, Ames, IA 50010, USA

**Keywords:** *Campylobacter*, antimicrobial resistance, cattle, intestinal colonization, feces, bovine respiratory disease (BRD), fluoroquinolone/enrofloxacin treatment, genotyping, minimum inhibitory concentration

## Abstract

Fluoroquinolone (FQ) resistance in a major foodborne bacterial pathogen, *Campylobacter jejuni*, derived from cattle has recently become prevalent and poses a significant public health concern. However, the underlying factors for this increase are not entirely clear. To evaluate the effect of enrofloxacin treatment on FQ-resistance development in *C. jejuni*, 35 commercial calves were equally divided into five groups (Groups 1–5) and were orally inoculated with FQ-susceptible (FQ-S) *C. jejuni*. Eight days later, Groups 4 and 5 were challenged with *Mannheimia haemolytica* via a transtracheal route to induce a respiratory disease; after 8 days, Groups 2, 3, 4, and 5 were injected subcutaneously with enrofloxacin (7.5 mg/kg for Groups 2 and 4, and 12.5 mg/kg for Groups 3 and 5). Colonization levels by FQ-resistant (FQ-R) and FQ-S *Campylobacter* in rectal feces were determined via differential culture throughout the experiment. Before oral inoculation with *C. jejuni*, only five calves were naturally colonized by *Campylobacter*, four of which were also colonized by FQ-R *C. jejuni* (three in Group 1 and one in Group 3). Soon after the oral inoculation, almost all calves in the groups became stably colonized by FQ-S *C. jejuni* (~3–6 log_10_ CFU/g), except that the four calves that were pre-colonized before inoculation remained positive with both FQ-R and FQ-S *C. jejuni*. Following enrofloxacin administration, *C. jejuni* colonization declined sharply and rapidly in all treated groups to undetectable levels; however, the vast majority of the animals were recolonized by *C. jejuni* at comparable levels 72 h after the treatment. Notably, no FQ-R *C. jejuni* was detected in any of the calves that received enrofloxacin, regardless of the drug dose used or disease status of the animals. The lack of detection of FQ-R *C. jejuni* was likely due to the localized high concentration of the antibiotic in the intestine, which may have prevented the emergence of the FQ-R mutant. These findings indicate that single-dose enrofloxacin use in cattle poses a low risk for selection of de novo FQ-R mutants in *C. jejuni*.

## 1. Introduction

*Campylobacter* is among the leading causes of foodborne bacterial gastroenteritis worldwide [1,2]. In Europe, there were 229,213 confirmed cases of campylobacteriosis, with a notification rate of 65.6 per 100,000 population in 2015 [3]. According to the Centers for Disease Control and Prevention (CDC), there are 1.3 million foodborne disease cases caused by *Campylobacter* each year in the U.S. [4]. In addition to chicken meat, which is well recognized as a major source of campylobacteriosis [5], beef and dairy cattle also contribute significantly to human *Campylobacter* infections [6,7,8]. Humans can acquire *Campylobacter* from cattle through direct contact, ingestion of unpasteurized milk, and water contamination [9,10,11]. *Campylobacter* species, primarily *Campylobacter jejuni* and *Campylobacter coli*, can be commonly found in the gastrointestinal tract of healthy calves and adult cattle, usually without causing any clinical illness [12,13,14,15]. Surveillance studies in the U.S. revealed a fecal carriage rate of approximately 50%, with most installations (e.g., feedlots, farms, and herds) testing positive for *Campylobacter* [16,17]. A relatively recent study conducted by our research group found an overall prevalence rate of approximately 75% (mostly *C. jejuni*, but also including *C. coli*) in the feedlot cattle from multiple states in the U.S. [18].

For clinical treatment of *Campylobacter* infections in humans, fluoroquinolones (such as ciprofloxacin) and macrolides (such as azithromycin) are among the most commonly prescribed antibiotics, as indicated [19,20]. However, many *Campylobacter* strains have developed resistance to both classes of antimicrobials, particularly FQs, posing a threat to treatment efficacy in individual patients [21,22,23]. Because *Campylobacter* is commonly present in the gastrointestinal tract of cattle, the bacterium is inevitably and readily exposed to antibiotics that are used to treat and prevent infectious diseases caused by a variety of bacterial pathogens, such as bovine respiratory disease (BRD). The etiology of BRD includes stress factors (e.g., weaning, castration, dehorning, transport, commingling, poor ventilation, and high stocking density), which compromise the cattle immune system and render it prone to various viruses and bacteria, among which *Mannheimia haemolytica* is considered a major pathogen [24]. Since BRD is a highly significant and costly widespread condition in U.S. feedlots, calves that are at high risk of developing BRD are frequently treated with FQ antibiotics following their arrival at the feedlot as a preventative measure [25,26,27]. Approximately 43% of U.S. feedlots reported treatment with an FQ antibiotic in roughly 42% of cattle with BRD, according to the Feedlot 2011 National Animal Health Monitoring System (NAHMS) study [28].

Among the FQ antibiotics, enrofloxacin is licensed as an injectable solution for use as both therapeutic (in sick animals) and metaphylaxis (in healthy animals at high risk of BRD development) for treatment of BRD associated with *M. haemolytica*, and other bacterial pathogens [26,27]. In the U.S., enrofloxacin can only be used in cattle production (e.g., beef cattle and non-lactating dairy cattle) with a veterinary prescription, and the extra-label use in food-producing animals is banned [29,30]. As a subcutaneous injection, it is used either as a single dose of 7.5 or 12.5 mg/kg of body weight for both therapeutic and metaphylactic purposes, or as a multi-day therapy at 2.5 or 5 mg/kg first and additional doses at 24 h, 48 h, and 72 h after the initial dose [31,32,33].

Over the past decade, FQ-resistance in *Campylobacter* isolates of cattle origin has been steadily increasing. In Northern Spain, the prevalence of FQ (ciprofloxacin)-resistant *C. jejuni* originated from beef cattle was found to have almost doubled in about 10 years (32% in 2003–2005 vs. 62% in 2014–2016) [34,35]. Between 2010 and 2011, a high prevalence of *Campylobacter* resistant to enrofloxacin was found in beef cattle in Japan, with 40% resistance in *C. jejuni* and 66.7% in *C. coli* [36]. In 2008, a slaughterhouse survey in the U.S. found that a significant percentage of *C. jejuni* (27.3%) and *C. coli* (49.2%) from various types of cattle production (e.g., feedlot cattle and adult animals) was resistant to ciprofloxacin [37]. Between 2012 and 2013, our study also found a high level of ciprofloxacin resistance in *Campylobacter* isolates (35.4% in *C. jejuni* and 74.4% in *C. coli*) collected from 35 feedlots located in five different U.S. states [18]. In contrast, the FQ resistance rate in *Campylobacter* isolates obtained from the 1999 Feedlot NAHMS national survey was found to be only 2.9% in the U.S. [28].

In an attempt to begin to better understand the driving factors responsible for the ongoing rise seen in FQ-resistance in bovine *Campylobacter*, we recently carried out a series of experiments to determine the effect of different FQ antibiotic treatment regimens on resistance development in both healthy and BRD-induced cattle. In the first study [38], we showed that a single dose of danofloxacin treatment had a minimal or no role in the emergence of de novo FQ-resistance in calves experimentally inoculated with FQ-susceptible *C. jejuni* strains. The current study is an expansion of the previous work and conducted to determine the effect of two different single doses of another commonly used FQ antibiotic (i.e., enrofloxacin) for BRD control on resistance development in *C. jejuni* in cattle. Our hypothesis is that different FQ treatment regimens (e.g., danofloxacin vs. enrofloxacin, and low vs. high dose) and the host status (healthy vs. diseased) may have different outcomes in FQ-resistance development in *Campylobacter*. Therefore, it is important to evaluate the effect of each of these variables individually for selection of the most appropriate treatment options for both an effective BRD control and antimicrobial stewardship program in the feedlot.

## 2. Results

### 2.1. Campylobacter Status of Calves Prior to Challenge

The summary of the main experimental procedures conducted in this study is presented in Table 1.

Culturing of rectal feces showed that most calves (30/35; 86%) were free of natural *Campylobacter* colonization before experimental inoculation with laboratory strains (days post-inoculation (DPI) −3 and 0 in Figure 1a,c,e,g,i). Interestingly, most of the colonized calves (*n* = 3) were included in the control group (Group 1), and Group 3 and Group 4 among the treatment groups each had one colonized calf only. Differential culture plating further revealed that FQ-resistant *C. jejuni* was present in all of the four colonized calves in Group 1 and Group 3, while the single colonized calf in Group 4 did not yield any resistant colonies (DPI −3 and 0 in Figure 1b,d,f,h,j). The relative percentage of FQ-resistant *C. jejuni* colonies in comparison to the total (susceptible and resistant) *C. jejuni* population in colonized animals was 43% in Group 1, and 82% in Group 3 (DPIs -3 and/or 0 in Figure 2a). These results indicated that most calves were free of *Campylobacter* colonization prior to experimental inoculation with the laboratory strains in the current study.

### 2.2. Bovine Respiratory Disease Induction

Since treating calves with BRD is one of the principal uses of enrofloxacin, we chose to mimic this in Group 4 and Group 5 to determine if the concomitant disease would influence FQ resistance development in *C. jejuni,* which could concurrently be present in the intestine as a commensal bacterium in cattle. Both groups of calves were inoculated with *M. haemolytica* and then observed for the BRD signs for a week using a scoring method established previously [39]. As expected, no signs of BRD prior to the inoculation were observed in any of the calves. At necropsy, two/seven calves in Group 4 and four/seven calves in Group 5 had characteristic lung lesions, including consolidation, rough surface, hyperemia, whereas none of the calves in the other three groups presented lesions of such a kind. In agreement, *M. haemolytica* was isolated from all six of the diseased lungs in the BRD-induced groups; none of the lungs in the other three groups were culture positive. Consequently, BRD-induction was considered mild to moderate in Group 4 and Group 5 while the other three groups were accepted to be free of BRD as judged by the combination of clinical, pathological, and culture results [39,40].

### 2.3. Experimental Inoculation of Calves with FQ-Susceptible C. jejuni Resulted in Effective Intestinal Colonization

Soon after (two days) the oral challenge with FQ-susceptible *C. jejuni* strains, all but two calves (33/35; 94%) became colonized by *C. jejuni* as measured by fecal shedding (DPI 2, Figure 1a,c,e,g,i). Differential plating showed that the colonization in all (*n* = 28) of the previously non-colonized calves (*n* = 30) was all by FQ-susceptible strains (Figure 1). Similar to the pre-inoculation period, four calves remained colonized by FQ-resistant *C. jejuni* (Figure 1b,f) even though the average percentage of FQ-resistant *C. jejuni* isolates compared to the total *C. jejuni* population declined substantially in these animals (DPI 2, Group 1 and Group 3, Figure 2a). Despite minor fluctuations, these observations remained quite consistent through DPIs 5–16, with all of the calves being colonized at some point during this sampling period (Figure 1).

### 2.4. Enrofloxacin Treatment Did Not Induce FQ-Resistance Development in the Intestine of Calves Colonized with FQ-Susceptible C. jejuni

Calves in the four treatment groups were injected subcutaneously with a single dose of enrofloxacin (7.5 mg/kg in Group 2 and Group 4; 12.5 mg/kg in Group 3 and Group 5) on day 16 following the oral administration of FQ-susceptible *C. jejuni* (DPI 16; Figure 1 and Figure 2). Enrofloxacin administration resulted in a very sharp and rapid decline in the number of calves colonized by *C. jejuni* in all four groups (i.e., from 27/28 pre-injection on DPI 16 to 1/28 post-injection on DPI 17; Figure 1c,e,g,i). However, this decrease was only transient and the vast majority of animals (24/28) were recolonized by *C. jejuni* on DPI 19. Similar trends were observed on DPI 21 and all of the animals remained colonized at the end of the experiment on DPI 23 at levels comparable to the pre-injection period (Figure 1c,e,g,i). Remarkably, as shown by differential plating, all of the *C. jejuni* populations in all of the colonized animals were FQ-susceptible; i.e., no FQ-resistant *C. jejuni* colonies were detected at all at any sampling point following the enrofloxacin injection (Figure 1 and Figure 2a). Of note, both the percentage of colonized animals and the level of colonization by total as well as FQ-resistant *C. jejuni* in Group 1 (not treated with enrofloxacin) overall remained stable during most of the study (Figure 1a,b).

### 2.5. Antimicrobial Susceptibility Profiles of C. jejuni Strains from Calves

In order to corroborate the results obtained by differential culture, ciprofloxacin MICs of *C. jejuni* isolates were determined using the Sensititre panel (a total of 217 isolates; one isolate from each positive calf on all of the sampling days was collected for this testing). The MIC test showed an overall high level of agreement in FQ susceptibility/resistance profiles of the isolates with those obtained by differential plating, further confirming that no FQ-resistant *C. jejuni* was detected following the enrofloxacin injection on DPI 16 in Group 2, Group 3, Group 4 and Group 5 (Figure 2a,b). Furthermore, the MIC testing revealed that there was no notable difference between the ciprofloxacin MICs of *C. jejuni* isolates collected prior to the enrofloxacin injection (MIC_90_ = 0.12; range 0.015–2; DPIs 2–16) and post-injection (MIC_90_ = 0.12; range 0.015–0.5; DPIs 17–23) (partly seen in Table 2). The Sensititre panel also indicated that nalidixic acid (a quinolone antibiotic) had MIC values and trends comparable to those of ciprofloxacin. In addition, the *C. jejuni* isolates showed a significant level of tetracycline resistance (77%). All other antibiotics included in the susceptibility testing (i.e., azithromycin, clindamycin, erythromycin, florfenicol, and gentamicin) had low MIC levels, which did not experience any substantial changes during the trial (results not shown).

### 2.6. Dynamics of C. jejuni Population throughout the Study

Genotyping was done to monitor the overall dynamic changes in the *C. jejuni* population in response to the major experimental procedures throughout the study (e.g., oral inoculation with *C. jejuni* and subcutaneous injection with enrofloxacin) (one isolate originating from each positive calf on all of the sampling days was included for testing; 217 total isolates). Overall, PFGE typing generated 11 unique macrorestriction profiles (designated genotypes a–k).

The composition of genotypes obtained during different periods of the study, including the acclimatization, post-inoculation with FQ-susceptible *C. jejuni* (prior to enrofloxacin administration), and post-injection with enrofloxacin, as well as the ciprofloxacin susceptibility phenotypes of isolates in Group 2, Group 3, Group 4, and Group 5, is shown in Table 2. Of the three isolates obtained prior to inoculation (DPI −3 and 0) from *Campylobacter*-positive calves, two were of the same genotype (j, an inoculum strain, ciprofloxacin-susceptible, Figure 1g), and the other had a different genotype (h, ciprofloxacin-resistant, Figure 1e,f). Following oral inoculation with FQ-susceptible laboratory *C. jejuni* strains (DPI 2–16; pre-injection with enrofloxacin), the number of genotypes increased from two to five (including both of the inoculum and three newly detected strains). Genotype a (including 43 isolates, all ciprofloxacin-susceptible) was the predominant genotype, followed by genotype c (*n* = 20 all ciprofloxacin-susceptible), genotype k (an inoculum strain, *n* = 14 all ciprofloxacin-susceptible), genotype j (an inoculum strain, *n* = 7 all ciprofloxacin-susceptible), and genotype b (*n* = 1 ciprofloxacin-susceptible). After the enrofloxacin injection (DPI 17–23), six genotypes (including all the five genotypes found pre-injection) were detected. Genotype c became the most prevalent (*n* = 30 isolates, all ciprofloxacin-susceptible), followed closely by the two inoculum strains (genotypes j and k, all ciprofloxacin-susceptible). The remaining seven isolates were represented by three genotypes (a, b, i) and were all ciprofloxacin-susceptible.

PFGE profiles and ciprofloxacin susceptibility phenotypes of the *C. jejuni* strains isolated from calves in Group 1 (no enrofloxacin given) are depicted in Table 3. Prior to inoculation with the laboratory strains of *C. jejuni,* genotype d (*n* = 2 ciprofloxacin-susceptible), genotype e (*n* = 2 ciprofloxacin-resistant), and genotype f (*n* = 2 ciprofloxacin-susceptible) were detected. The number of genotypes increased substantially post-inoculation (from three to nine, including both the inoculum and five newly detected strains), with genotype a being the predominant (*n* = 17–all but three ciprofloxacin-susceptible), followed by genotype c (*n* = 13–all but one ciprofloxacin-susceptible). The remaining 16 isolates were represented by seven genotypes and were mostly ciprofloxacin susceptible (Table 3).

## 3. Discussion

Cattle are a major reservoir of *Campylobacter*, including FQ-resistant isolates [18,41,42]. Therefore, it is crucial to assess if the FQ treatment regimen can be optimized to minimize the selection pressure on *Campylobacter* and the magnitude of FQ-resistance in cattle. As shown in our recent study [38], single dose subcutaneous danofloxacin (an FQ antibiotic) treatment did not cause any quantifiable level of de novo resistance in FQ-susceptible *C. jejuni* strains in the intestine of either healthy or BRD-induced calves. In that study, the vast majority of calves were naturally colonized (an un-ideal situation for experimental purposes) by FQ-resistant *C. jejuni* prior to the experimental inoculation, which experienced a sharp but brief spike after the FQ injection. In contrast, the calves used in the current study were mostly free of natural *Campylobacter* colonization at the procurement, which was highly desirable to perform the principal goal of this investigation since finding (FQ-resistant) *Campylobacter*-free calves from commercial sources can be quite a difficult task [38]. A key finding of the present study is that single dose subcutaneous enrofloxacin treatment (regardless of the dose administered) did not result in the emergence of FQ resistance in FQ-susceptible *C. jejuni* colonizing the intestine of calves (irrespective of the BRD status). Additionally, a noteworthy observation is that enrofloxacin treatment did not eliminate the pre-existing FQ-susceptible *C. jejuni* population in the intestine of calves; instead, it merely caused a transient, yet sharp, decline in the colonization level.

As can be seen from Figure 1, oral inoculation with FQ-susceptible *C. jejuni* strains resulted in intestinal colonization in almost all of the previously non-colonized calves (28/30) with FQ-susceptible *C. jejuni*, indicating the effectiveness of the challenge approach utilized in this study. Although the colonization remained quite stable and consistent during the next two-week period, the subcutaneous enrofloxacin injection performed on DPI 16 led to *Campylobacter* being undetected in all but one of the calves soon after (i.e., within 24 h) the injection (Figure 1c,e,g,i). However, on DPI 19 (i.e., 72 h after the enrofloxacin injection), both the number of colonized calves and the magnitude of colonization (CFU/g feces) returned to the levels comparable to the pre-injection values and remained as such for the next 4 days until the end of the experiment. In line with these observations, fecal concentrations of enrofloxacin and its active metabolite ciprofloxacin were found to be at the peak levels (~20–40 µg/g) during the 12–24 h period after the antibiotic injection, and almost totally eliminated 48 h after the injection in all four treatment groups regardless of the dose administered or BRD status of the calves [43]. Interestingly and importantly, as shown by differential plating and MIC determination (Figure 1 and Figure 2), the re-establishment of the colonization observed soon after the enrofloxacin injection (DPI 19 and beyond) in all of the calves in all four treatment groups was by FQ-susceptible *C. jejuni*. This finding indicated that a single dose subcutaneous enrofloxacin treatment, as employed in the current study, did not result in any detectable level of FQ-resistance development from FQ-susceptible *C. jejuni* inhabiting the intestine of calves. As mentioned above, highly similar results were obtained with another FQ-antibiotic (danofloxacin) treatment of calves in a recent experimental study conducted by our research group [38], as well as in a field study in which feedlot cattle were treated with a single subcutaneous dose (7.5 mg/kg) of enrofloxacin for metaphylactic purposes [44].

In stark contrast to the aforementioned findings observed in cattle, FQ-resistant *Campylobacter* emerges rapidly from FQ-susceptible *Campylobacter* strains colonizing the chicken intestine and remains as the predominant population long after the completion of treatment with different FQs (e.g., enrofloxacin, sarafloxacin, or difloxacin; usually given in drinking water for multiple consecutive days), as shown in both experimental and field studies [45,46,47,48,49]. Even though the exact reason for these totally distinct outcomes cannot be definitively stated from the currently available data, several plausible explanations can be given. Firstly, the differences in the antibiotic treatment regimes (e.g., oral vs. parenteral, single dose vs. multiple doses, etc.) employed in different host species may account for the different effects in chickens vs. cattle. Different regimes are likely to result in different drug concentrations in the intestine and associated alterations in the gut microbiota. Secondly, the distinct anatomic/physiologic features to be found in the gastrointestinal tract of chickens and cattle (avian vs. ruminant digestive systems) may influence the interactions between the residing bacteria and their response to various insults such as antibiotics. Lastly, differences in *Campylobacter* loads (CFU/g) inhabiting the intestine of different host species could be a key determinant. The magnitude of colonization by *Campylobacter* in the chicken intestinal tract is typically very robust, reaching up to 9.0 log_10_ CFU/g in the ceca [50]. However, the colonization level by *Campylobacter* in the cattle intestine is usually much lower than in chickens (i.e., 2 to 5 log10 CFU/g feces) [38,51]. Similarly, in the current study, the mean colonization level (as measured in freshly collected rectal feces) of *C. jejuni* ranged between 4.1 and 4.6 log10 CFU/g feces although this value was close to 7 log10 CFU/g prior to the enrofloxacin injection in a few individual calves (partly shown in Figure 1). We previously showed that both the frequency (~10^6^–10^8^) of emergence of spontaneous FQ-resistant (MIC ≥ 4 µg/mL) mutants and the development of FQ resistance under antibiotic treatment in *C. jejuni* were influenced by the magnitude of the selection pressure (antibiotic concentration; 0.625–4 µg/mL) and the initial bacterial cell density using in vitro experiments [52,53]. The results from those studies suggested that the successful development of FQ resistance in *C. jejuni* during antibiotic exposure required an initial cell density of at least 6 log10 CFU/mL and a FQ antibiotic concentration of at least 0.625 µg/mL (or 5X MIC of the strain used). Given that the highest level of *C. jejuni* detected in pre-treatment rectal feces of this study was around 6–7 log10 CFU/g (likely higher in the intestine) in a few calves prior to the enrofloxacin injection in each of the four treatments groups (Figure 1c,e,g,i) and that the concentrations of enrofloxacin (and its metabolite ciprofloxacin) were far above 4 µg/g feces for at least 24 h following the antibiotic injection [43], it is reasonable to assume that spontaneous FQ-resistant mutants would have been selected if they existed in the calves. Obviously, this was not the case as development of FQ-resistant *C. jejuni* was not detected in this study.

The pharmacokinetic data provide important information about the localized FQ concentrations in the intestines of calves, which could explain why no FQ-resistant mutants were detected in this study [43]. As reported in previous publications, the typical MICs of ciprofloxacin in FQ-resistant *C. jejuni* is between 4–16 µg/mL [22,46,47,53,54]. The peak concentration of enrofloxacin was found to be around 2–4 µg/mL in the intestines of broiler chickens during a standard multi-dose enrofloxacin water treatment in a previous study, in which FQ-resistant *C. jejuni* developed soon after the start of enrofloxacin treatment [54]. However, the drug concentration in the rectal feces of calves examined in the current study is much higher (median: 38–54 µg/g feces for enrofloxacin and 18–21 µg/g feces for ciprofloxacin within 12 h of the enrofloxacin injection and remaining at comparable levels by 24 h post-injection) [43]. Such a high level of antibiotic selection pressure may have reached above the mutant selection window [55] and thus prevented the emergence of FQ-resistant mutants in the intestines of calves. Similarly, a multiple dose regimen (5 mg/kg, 3 consecutive days) of enrofloxacin resulted in a more persistent and significantly higher level of increase in the MIC of FQ-susceptible *E. coli* isolates than its single dose (12.5 mg/kg) in the intestine of 6-month-old calves following a subcutaneous injection [56]. These results suggest that the injectable FQ antibiotics, which result in a high antibiotic concentration in the intestine, pose a low risk for de novo development of FQ resistance from a FQ-susceptible population.

In this study, PFGE (as well as MLST) was used to monitor the dynamic changes in the *C. jejuni* population in response to major experimental procedures performed throughout the study. Interestingly, both the oral inoculation with FQ-susceptible *C. jejuni* strains and the enrofloxacin injection were followed by notable increases in the number of different genotypes detected, including not only the inoculum strains but also newly detected genotypes (Table 2 and Table 3). Overall, two new genotypes (a and c, which were not found pre-inoculation) and the two inoculum genotypes (j and k) dominated the post-inoculum and post-injection periods. Additionally, it is noteworthy to point out the remarkable shift observed in genotype distribution after the antibiotic injection, where genotype a declined sharply and genotypes c and j became predominant (Table 2). These findings illustrate the highly dynamic nature of *Campylobacter* colonization at the population level and suggest that certain strains (e.g., genotypes a and c) may be better adapted to cattle host and respond differently (e.g., genotypes a vs. c) to various disturbances in the intestinal tract. Indeed, we and other investigators showed the relatively common occurrence of genotype a (ST-929) and genotype c (ST-61) in cattle in previous studies [38,57,58].

## 4. Materials and Methods

### 4.1. Animals and Study Design

Thirty-five male Holstein calves were purchased from a farm in Wisconsin in May 2019. The animals were between 3 and 4 months old and weighed. The calves were free of known antibiotic exposure and showed no overt clinical disease on arrival. Upon their arrival at the animal facility, veterinarians from Iowa State University (ISU) visually examined them for any indications of sickness as described previously [38]. During the study, none of the animals experienced any serious health issues that would have required additional treatment. The animals were randomly assigned into groups (n = seven calves per group) and each individual was assigned a unique identification number. Group 1 was orally inoculated with *C. jejuni* and received no antibiotic treatment. This group served as a non-antibiotic-treated control. Groups 2 and 3 were first inoculated with *C. jejuni* and 16 days later were treated with a single dose of enrofloxacin by subcutaneous administration: 7.5 mg/kg for Group 2 and 12.5 mg/kg for Group 3. Groups 4 and 5 were first inoculated with *C. jejuni*, 8 days later challenged with *Mannheimia haemolytica*, and another 8 days later treated with a single dose of enrofloxacin: 7.5 mg/kg for Group 4 and 12.5 mg/kg for Group 5. Appendix A provides pertinent details on the bacterial isolates utilized in the challenge studies. During the study, all animals were housed in the laboratory animal resources facility at Iowa State University (ISU) under biosafety level 2 containment. Feed used for the animals were mixed grass hay and a premixed calf starter (Heartland Co-op, Des Moines, IA, USA) and water was given ad libitum. This animal study was reviewed and approved by the Institutional Animal Care and Use Committee (IACUC-18-372) at ISU.

### 4.2. Inoculation of C. jejuni

All the calves included in the study were inoculated with a mixture of two different FQ-susceptible *C. jejuni* strains (ciprofloxacin MIC = 0.125 μg/mL) as previously described [38]. The strains used for inoculation were IA-6-FC-30 and MO-2-FC-25, which originated from feedlot cattle and belonged to different PFGE/MLST subtypes [59]. Each calf was orally inoculated with 60 mL (~4 × 10^9^ CFU/mL) of the strain cocktail directly into the rumen using an esophageal tube. Prior to being used in this experiment, the two strains were tested to be highly motile on semi-solid agar as described elsewhere [60].

### 4.3. Inoculation with Mannheimia haemolytica

Eight days after *C. jejuni* inoculation, calves in Groups 4 and 5 were inoculated with *M. haemolytica* to induce BRD. The challenge was done by transtracheal injection using a catheter [38,61]. The *M. haemolytica* strain used in this study and preparation of the inoculum were described in our previous publication [38]. Each inoculum included 20 mL of *M. haemolytica* culture (~5.0 × 10^8^ CFU/mL). After the challenge, the animals were observed and monitored for BRD symptoms, such as fever, depression, ocular and nasal discharges, ear droop or head tilting, cough, and changes in respiration, feeding, and ambulation. The calves were clinically categorized as BRD-positive or BRD-negative based on a scoring system as described elsewhere [39].

### 4.4. Enrofloxacin Injection

Sixteen days after *C. jejuni* inoculation, the calves in Groups 2, 3, 4, and 5 were subcutaneously (sc) injected in the neck with a single dose of enrofloxacin (BAYTRIL^TM^ 100, Bayer Animal Health, Shawnee Mission, KS, USA). The dose for Groups 2 and 4 was 7.5 mg/kg body weight, while the dose for Groups 3 and 5 was 12.5 mg/kg body weight.

### 4.5. Collection of Fecal Samples

Fecal samples were collected as described in our previous study [38] on DPI −3, 0, 2, 5, 8, 16, 17, 19, 21, and 23. The day of oral inoculation with *C. jejuni* was regarded as day 0 and fecal samples collected on that day were done prior to *C. jejuni* inoculation.

### 4.6. Bacterial Isolation and Identification

For bacterial isolation, fecal samples were serially diluted in MH broth and the dilutions were plated onto MH agar plates supplemented with Preston *Campylobacter*-selective supplement (SR117E; Oxoid) and *Campylobacter* growth supplement (SR084E; Oxoid, Basingstoke, UK). For the enumeration of FQ-resistant *C. jejuni*, the MH agar media were added with 4 µg/mL ciprofloxacin. All plates were incubated under microaerobic conditions at 42 °C for 48 h. Two *Campylobacter*-like colonies were randomly picked for each sample from the MH agar plates devoid of ciprofloxacin. The colonies were sub-cultured onto fresh plain MH agar plates to produce pure cultures. In order to confirm the *Campylobacter* status of the calves before experimental inoculation, enrichment culture was performed on fecal samples obtained before DPI 0. Although it is not quantitative, the enrichment approach is more sensitive than direct plating when the colonization level is low, as previously mentioned [38]. All of the purified isolates were confirmed and identified at the species level using MALDI-TOF mass spectrometry (Bruker Daltonik, Billerica, MA, USA).

### 4.7. Pulsed-Field Gel Electrophoresis (PFGE)

PFGE analysis of *C. jejuni* isolates was done using the *Sma*I restriction enzyme and was performed as described in our previous publication [38].

### 4.8. Multilocus Sequence Typing (MLST)

Representative isolates of different PFGE types were also analyzed by MLST as described previously [38].

### 4.9. Minimum Inhibitory Concentration (MIC) Determination

The MICs of various antibiotics against *C. jejuni* isolates were determined following the methods described in our previous study [38].

### 4.10. Necropsy

At the end of the study, calves were euthanized by using a penetrating captive bolt gun. Necropsy and collection of lung samples for *M. haemolytica* culture and identification were performed as described previously [38].

### 4.11. Statistical Analysis

Statistical analysis was conducted as described in our previous study [38].

## 5. Conclusions

Findings from the current study clearly indicated that a single injection of two different doses of enrofloxacin was not associated with any measurable level of FQ-resistance development in the intestine of calves colonized with FQ-susceptible *C. jejuni*. This is in agreement with the results from our previous study [38], where we showed that a single-dose danofloxacin treatment did not lead to the selection of de novo FQ-R mutants in susceptible strains in calves that were colonized with a mixture of both FQ-susceptible and FQ-resistant *C. jejuni*. In agreement with our observations, a recent field study also found no evidence of selection of FQ-resistance in *Campylobacter* in feedlot cattle at risk of BRD development following a single-dose enrofloxacin injection [44]. Altogether, the data obtained from independent studies so far strongly suggest that single-dose use of FQ antibiotics (the most common form used in U.S. feedlots) for BRD metaphylaxis or treatment poses a low risk for selecting FQ-resistant *Campylobacter* in the intestines of cattle. This is likely due to the relatively short but high concentrations of FQ antibiotics reached in the intestines of cattle following parenteral administration [43], which creates an unfavorable environment for the emergence of de novo FQ-resistant mutants in *C. jejuni*. However, there may be some residual antibiotic concentration present in the intestine after the treatment. Whether the residual concentration may help to serve as a selection force for pre-existing FQ-resistant mutants is unknown and remains to be examined. Since both danofloxacin and enrofloxacin also have multi-dose treatment regimens (with lower doses administered) approved for BRD treatment in cattle, it would be valuable to ascertain the effect of such uses (though less commonly practiced) on FQ-resistance development in *Campylobacter* to better inform future policy decisions.

## Figures and Tables

**Figure 1 antibiotics-11-01407-f001:**
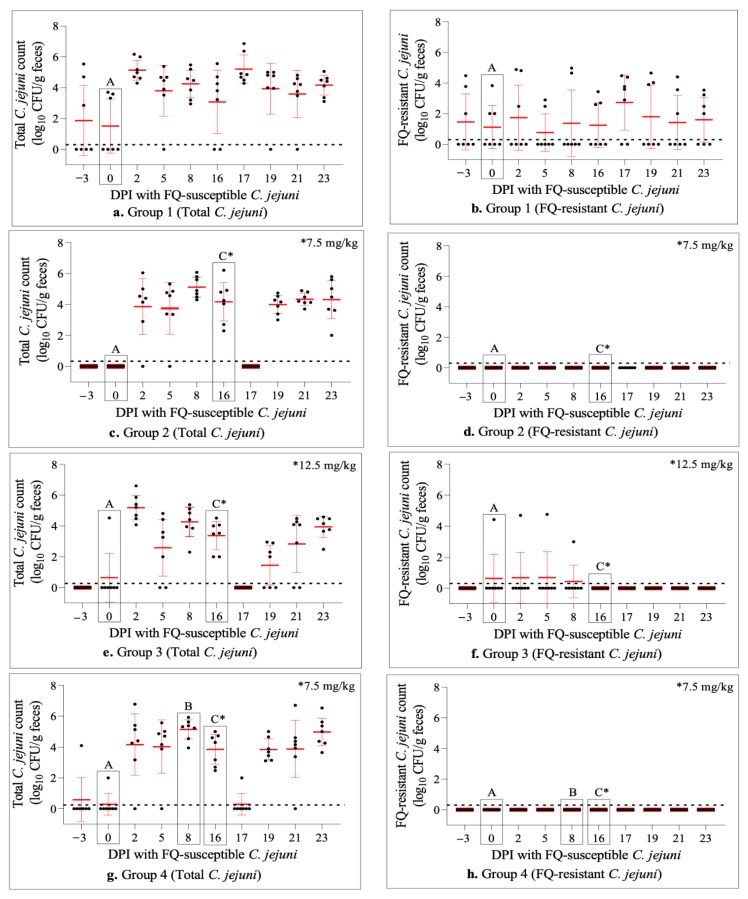
Dot plot analysis of *Campylobacter* colonization level (log_10_ CFU/g feces) in cattle of groups 1–5. Panels (**a**,**c**,**e**,**g**,**i**) represent total *C. jejuni* CFUs (both susceptible and resistant), while (**b**,**d**,**f**,**h**,**j**) indicate FQ-resistant *C. jejuni* population. Each dot indicates the CFU number from an individual calf. The horizontal red bars depict the mean colonization levels, while the vertical red lines show the 95% confidence interval. Letter A designates the time when all calves were orally inoculated with FQ-susceptible *C. jejuni*, letter B denotes the time when *M. haemolytica* was given to the calves in Groups 4 and 5, and letter C depicts the day on which groups 2, 3, 4, and 5 received subcutaneous injection of enrofloxacin. The culture’s detection limit was approximately 100 CFU/g of feces (shown as dotted black lines over *x*-axis). DPI stands for day after inoculation.

**Figure 2 antibiotics-11-01407-f002:**
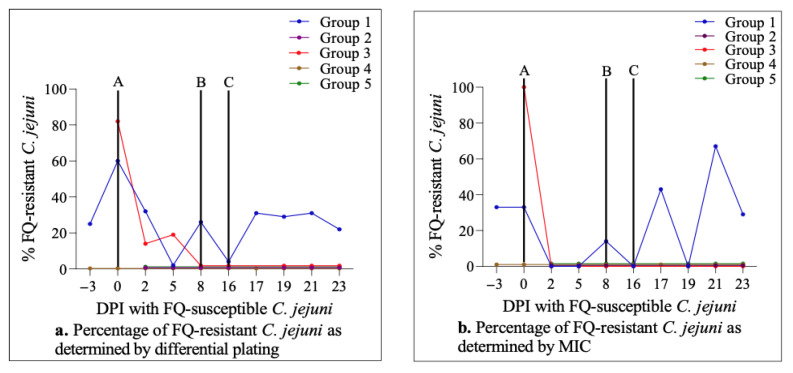
The proportions of FQ-resistant *C. jejuni* colonies in colonized calves as determined by differential plating (**a**) and MIC assays (**b**). The 5 groups are represented by lines of different colors. See the legend of Figure 1 for letters A, B, and C.

**Table 1 antibiotics-11-01407-t001:** Experimental design.

Groups	Inoculation with FQ-S *C. jejuni **	Challenge with *M. haemolytica ^#^*	Enrofloxacin Injection (Dose mg/kg) ^§^
1	Yes	No	No
2	Yes	No	Yes (7.5)
3	Yes	No	Yes (12.5)
4	Yes	Yes	Yes (7.5)
5	Yes	Yes	Yes (12.5)

* On DPI (days post-inoculation) 0, all the calves were orally challenged with FQ-S *C. jejuni* following acclimatization for four days. ^#^ On DPI 8, the calves in Groups 4 and 5 were challenged with *M. haemolytica* using the transtracheal route. ^§^ On DPI 16, enrofloxacin was given subcutaneously to calves (7.5 mg/kg in Group 2 and Group 4; 12.5 mg/kg in Group 3 and Group 5). Collection of rectal feces was done during the entire study, which ended on DPI 23.

**Table 2 antibiotics-11-01407-t002:** PFGE profiles and ciprofloxacin MICs of *C. jejuni* isolates obtained from calves in groups administered with enrofloxacin (Groups 2, 3, 4, and 5) ^#^.

Pre-Inoculation (DPI –3 and 0), *n* = 3			Post-Inoculation (DPI 2–16), *n* = 85			Post-Injection (DPI 17–23), *n* = 77		
Genotype *	CIP ^§^	MIC *	Genotype	CIP	MIC	Genotype	CIP	MIC
a (0)	---	---	a (43)	S	0.12 (32); 0.25 (11)	a (3)	S	0.12 (3);
b (0)	---	---	b (1)	S	0.12 (1)	b (3)	S	0.12 (2); 0.5 (1)
c (0)	---	---	c (20)	S	0.015 (1); 0.12 (12); 0.25 (6); 0.5 (1)	c (30)	S	0.06 (2); 0.12 (21); 0.25 (5); 0.5 (2);
d (0)	---	---	d (0)	---	---	d (0)	---	---
e (0)	---	---	e (0)	---	---	e (0)	---	---
f (0)	---	---	f (0)	---	---	f (0)	---	---
g (0)	---	---	g (0)	---	---	g (0)	---	---
h (1)	R	8 (1)	h (0)	---	---	h (0)	---	---
i (0)	---	---	i (0)	---	---	i (1)	S	0.12 (1)
j (2) ^¶^	S	0.12 (2)	j (7)	S	0.06 (2); 0.12 (5)	j (24)	S	0.015 (1); 0.06 (10); 0.12 (12); 0.25 (1)
k (0) ^¶^	---	---	k (14)	S	0.12 (10); 0.25 (3); 2 (1)	k (16)	S	0.12 (12); 0.25 (4)

^#^ DPI −3 and 0 are isolates collected before the inoculation with laboratory strains of FQ-susceptible *C. jejuni.* DPI 2–16 include isolates collected between the time of the *C. jejuni* inoculation and the time right before the enrofloxacin administration. Isolates obtained following injection of enrofloxacin are included in DPI 17–23. “*n*” stands for the quantity of isolates examined throughout each time interval. * Lowercase letters represent different genotypes (macrorestriction pattern). The numbers of isolates and ciprofloxacin MIC with a specific genotype are shown by numbers in parentheses. ^§^ Ciprofloxacin susceptibility phenotype; R stands for resistant (MIC ≥ 4) and S stands for susceptible (MIC ≤ 2). ^¶^ Genotypes of the strains that were employed as the inoculum.

**Table 3 antibiotics-11-01407-t003:** PFGE profiles and ciprofloxacin MICs of *C. jejuni* isolates obtained from calves that did not receive enrofloxacin (Group 1) ^#^.

Pre-Inoculation (DPI –3 and 0), *n* = 6			Post-Inoculation (DPI 2–23), *n* = 46		
Genotype *	CIP ^§^	MIC *	Genotype	CIP	MIC
a (0)	---	---	a (17)	S/R	0.12 (13); 0.25 (1); 4 (1); 8 (2)
b (0)	---	---	b (2)	S	0.06 (1); 0.12 (1)
c (0)	---	---	c (13)	S/R	0.12 (10); 0.25 (2); 4 (1)
d (2)	S	0.06 (2)	d (2)	S	0.12 (1); 0.25 (1)
e (2)	R	4 (1); 8 (1)	e (5)	R	4 (1); 8 (4)
f (2)	S	0.06 (1); 0.12 (1)	f (0)	---	---
g (0)	---	---	g (1)	R	8 (1)
h (0)	---	---	h (1)	S	0.25 (1)
i (0)	---	---	i (0)	---	---
j (0) ^¶^	---	---	j (3)	S	0.06 (3)
k (0) ^¶^	---	---	k (2)	S	0.12 (2)

^#^ DPI −3 and 0 represent isolates collected before the inoculation with laboratory strains of FQ-susceptible *C. jejuni.* Isolates acquired between post-*C. jejuni* inoculation and necropsy are represented by DPI 2–23. “*n*” stands for the quantity of isolates examined during each time interval. * Lowercase alphabetical letters indicate distinct genotypes (macrorestriction pattern). The number of isolates and ciprofloxacin MICs with a specific genotype are shown by numbers in parentheses. ^§^ Ciprofloxacin susceptibility phenotype: R stands for resistant (MIC ≥ 4) and S stands for susceptible (MIC ≤ 2). ^¶^ The genotypes of strains that were employed as inoculum.

## Data Availability

Data are available upon request.

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
