# Peer review of "Influence of Single Dose Enrofloxacin Injection on Development of Fluoroquinolone Resistance in Campylobacter jejuni in Calves"

_antibiotics, 2022, doi:10.3390/antibiotics11101407_

Round 1
Reviewer 1 Report
The manuscript “Influence of Single Dose Enrofloxacin Injection on Development of Fluoroquinolone Resistance in Campylobacter jejuni in Calves” was the aim to determine the effect of two different single doses of FQ antibiotic (enrofloxacin) for BRD control on resistance development in C. jejuni in cattle. The manuscript is clear, it well conducted and the results are crearly presented. The scientific information is relevant.
Minor revision
- Line 352 and Line 412, “However, the drug concentration in the rectal feces of calves examined in the current study is much higher (median: 38-54 µg/g feces for enrofloxacin and…) ¿how authors measure drug concentration in feces? This methodology is not describe in the corresponding section.
- Results of MLST are not describe in the results section, only in line 438 (discussion section) authors mentioned that they detected two ST commonly describe for other authors.
Author Response
We greatly appreciate the highly constructive comments and tried to address each one of them as detailed below. We hope these changes would satisfactorily address the Reviewer’s suggestions and concerns.
Point 1: Line 352 and Line 412, “However, the drug concentration in the rectal feces of calves examined in the current study is much higher (median: 38-54 µg/g feces for enrofloxacin and…) How authors measure drug concentration in feces? This methodology is not describe in the corresponding section.
Response 1: The drug concentrations in the rectal feces of calves were measured by LC–MS/MS. This part of the study was recently published as a separate publication by our research group (Reference #44 in the current manuscript): “Beyi, A.F., Mochel, J.P., Magnin, G. et al. Comparisons of plasma and fecal pharmacokinetics of danofloxacin and enrofloxacin in healthy and Mannheimia haemolytica infected calves”. Sci Rep 12, 5107 (2022). https://doi.org/10.1038/s41598-022-08945-z.
Point 2: Results of MLST are not describe in the results section, only in line 438 (discussion section) authors mentioned that they detected two ST commonly describe for other authors.
Response 2: In this study, we primarily utilized PFGE to monitor the dynamic changes in C. jejuni population in response to major experimental procedures performed. Since PFGE is, in general, more discriminatory than MLST for C. jejuni, only the results for PFGE are described in details for the purpose of this study as it provided more informative data. MLST was performed only on a subset of isolates (the most common PFGE genotypes) to provide some global context for the genetic diversity observed. Beyond that, the utility of MLST is quite limited for the current study and does not really provide much critical/relevant information, and thus was not presented in detail here. However, we would be happy to provide more data on MLST if the Reviewer still feels that information would be valuable (probably as supplementary information).
Reviewer 2 Report
The authors study on "Influence of Single Dose Enrofloxacin Injection on Develop- 2 ment of Fluoroquinolone Resistance in Campylobacter jejuni in 3 Calves". The study results showed that a single injection of two different doses of enrofloxacin was not associated with any measurable level of FQ resistance development in the intestine of calves colonized with FQ-susceptible C. jejuni.
Overall, the results are very interesting, and the manuscript is written very well. However, there are some minor comments to be made before publication;
Line 468-469: "4.2. Oral Inoculation with Fluoroquinolone-Susceptible C. jejuni: The calves were orally inoculated via an esophageal tube with a mixture o...........................at the ISU animal facility" Is the mixture of two FQ-susceptible C. jejuni strains were inoculated into all of the calves ? please elaborate it.
The authors also need to add the ethical statement.
Author Response
We thank to the Reviewer for the highly constructive comments and tried to address each one of them as detailed below. We hope these changes would satisfactorily address the Reviewer’s suggestions and concerns.
Point 1: Overall, the results are very interesting, and the manuscript is written very well. However, there are some minor comments to be made before publication. Line 468-469: "4.2. Oral Inoculation with Fluoroquinolone-Susceptible C. jejuni: The calves were orally inoculated via an esophageal tube with a mixture o...........................at the ISU animal facility" Is the mixture of two FQ-susceptible C. jejuni strains were inoculated into all of the calves ? please elaborate it.
Response 1: Yes, the mixture of two FQ-susceptible C. jejuni strains were orally inoculated into all of the calves. We changed the phrase “The calves were orally inoculated…” to “All the calves included in the study were orally inoculated…” (line 469).
Point 2: The authors also need to add the ethical statement.
Response 2: We included the required information regarding the animal use in the manuscript (Material and Methods; Animals and Study Design): “All animals were cared for and handled according to the protocols and procedures approved prior to the start of the study by the Institutional Animal Care and Use Committee (IACUC-18-372) at ISU” (lines 464-466). If there is more pertinent/necessary information/statement on this subject is needed, we would be happy to provide that if the Journal Editorial Office would let us know.